# Endocrine Fibroblast Growth Factors in Relation to Stress Signaling

**DOI:** 10.3390/cells11030505

**Published:** 2022-02-01

**Authors:** Makoto Shimizu, Ryuichiro Sato

**Affiliations:** 1Nutri-Life Science Laboratory, Department of Applied Biological Chemistry, Graduate School of Agricultural and Life Sciences, University of Tokyo, Tokyo 113-8657, Japan; roysato@g.ecc.u-tokyo.ac.jp; 2Food Biochemistry Laboratory, Department of Applied Biological Chemistry, Graduate School of Agricultural and Life Sciences, University of Tokyo, Tokyo 113-8657, Japan

**Keywords:** FGF15/19, FGF21, FGF23, endocrine FGFs, stress signaling, ER stress, oxidative stress

## Abstract

Fibroblast growth factors (FGFs) play important roles in various growth signaling processes, including proliferation, development, and differentiation. Endocrine FGFs, i.e., atypical FGFs, including FGF15/19, FGF21, and FGF23, function as endocrine hormones that regulate energy metabolism. Nutritional status is known to regulate the expression of endocrine FGFs through nuclear hormone receptors. The increased expression of endocrine FGFs regulates energy metabolism processes, such as fatty acid metabolism and glucose metabolism. Recently, a relationship was found between the FGF19 subfamily and stress signaling during stresses such as endoplasmic reticulum stress and oxidative stress. This review focuses on endocrine FGFs and the recent progress in FGF studies in relation to stress signaling. In addition, the relevance of the stress–FGF pathway to disease and human health is discussed.

## 1. Introduction

Fibroblast growth factors (FGFs) are polypeptides that regulate various physiological functions, including growth, differentiation, development, wound healing, and energy metabolism [1,2]. FGFs activate intracellular signaling pathways by binding to cell-surface FGF receptors (FGFRs). FGFs contain a homologous core region of 120–130 amino acids arranged into 12 antiparallel β-strands, flanked by divergent amino- and carboxyl-termini. Sequence variation of the amino- and carboxyl-terminal tails usually accounts for the different biological functions of FGFs [3,4]. To date, twenty-two FGFs and four FGFRs have been identified in mammals. In invertebrates, two (*egl-17* and *let-756*) and three genes (*branchless*, *pyramus*, and *thisbe*) for FGFs have been reported in *Caenorhabditis elegans* and *Drosophila melanogaster*, respectively. These genes encode polypeptides with a core region similar to mammalian FGFs. FGFRs are also found in *Caenorhabditis elegans* (*egl-15*) and *Drosophila melanogaster* (*breathless* and *heartless*), suggesting that FGF/FGFR signaling is evolutionarily conserved [5,6,7,8,9]. In mammals, FGFs can be divided into seven subfamilies (FGF1, 4, 7, 8, 9, 11, and 19 subfamilies) based on their biological functions, sequence homology, and evolutionary relationships. Five subfamilies (FGF1, 4, 7, 8, and 9) are the autocrine/paracrine FGFs, which are mainly important for multiple developmental processes. The FGF11 subfamily is the intracellular FGFs. Unlike other FGF subfamilies, these FGFs are not secreted, but interact with cytosolic proteins to regulate intracellular signaling, such as ion channels. The FGF19 subfamily (FGF15/19, FGF21, and FGF23) is the endocrine FGF. In mammals, human FGF15 and mouse FGF19 are lacking, as human FGF19 and mouse FGF15 are orthologs based on comparative genomics (in this review, we describe them as “FGF15/19”). Most FGF members have a high affinity for heparan sulfate glycosaminoglycans in the extracellular matrix, which enables them to act in an autocrine and a paracrine manner. Unlike conventional FGFs, endocrine FGFs (i.e., FGF15/19, FGF21, and FGF23) have a weak affinity for heparan sulfate glycosaminoglycans; therefore, they require a single-pass transmembrane protein, Klotho (αKlotho, βKlotho, and lactase-like), as a coreceptor to allow binding to FGFRs. Upon binding to FGFs, FGFRs are activated by dimerization and autophosphorylation; subsequently, cytosolic substrates of FGFRs, such as the FGFR substrate 2α and mitogen-activated protein kinases, are activated [1]. αKlotho functions as a coreceptor for FGF23, whereas βKlotho serves as a coreceptor for both FGF15/19 and FGF21 [10,11,12,13,14,15]. Although FGFRs are broadly expressed, the expression of Klotho occurs only in specific tissues [16]. This allows endocrine FGFs to enter circulation and function as hormones. The biological functions of FGFs are classically considered to be related to development and differentiation. Recent studies, however, have found that FGF15/19 and FGF21 are important regulators of nutrient and energy metabolism, whereas FGF23 is vital for phosphate and vitamin D homeostasis [17,18]. Furthermore, endocrine FGFs may play certain roles in metabolic disorders; hence, these polypeptides have become attractive target molecules for the development of therapies [19].

The number of patients with metabolic abnormalities, such as obesity, diabetes mellitus, and hypertension, continues to increase worldwide. These disorders, which are caused by an imbalance in energy intake and expenditure, as well as an unhealthy lifestyle (e.g., a western-style diet and physical inactivity), are major risk factors for serious disease, including cardiovascular and cerebrovascular diseases. Therefore, lifestyle modification is important to prevent metabolic abnormalities [20,21]. In addition, the intake of functional food factors, such as bioactive food-derived molecules, is an attractive approach to preventing metabolic disorders [22,23].

Adaptation to a multitude of environmental stresses is essential for the survival of multicellular organisms. When cells are exposed to stress, intracellular stress signaling is activated to prevent stress-induced damage and to maintain cellular homeostasis. Physical and chemical stresses can include endoplasmic reticulum (ER) stress and oxidative stress. It is now known that endocrine FGFs play an important role in stress signaling as well as energy metabolism. In this review, we summarize the role of endocrine FGFs in the regulation of energy metabolism, and we detail their contribution to various stress signaling processes. In addition, we discuss how the stress–FGF pathway is relevant to disease and human health.

## 2. Endocrine FGFs

### 2.1. FGF15/19

Mouse FGF15 was initially found to be expressed in the developing nervous system [24], and human FGF19 was discovered following a homology-based search [25]. Although the amino acid sequences of mouse FGF15 and human FGF19 are dissimilar, comparative genomics has shown that their genes are orthologs [26]; thus, the two are referred to as “FGF15/19”. The primary source of FGF15/19 is the ileum, a distal part of the small intestine that absorbs bile acids [16]. Once released from the ileum, FGF15/19 travels to target tissues, including the liver (Figure 1). For binding to FGF receptors, FGF15/19 requires the presence of another transmembrane protein, βKlotho [11,27]. In the liver, FGF15/19 strongly represses the expression of *Cyp7a1*, a rate-limiting enzyme involved in bile acid biosynthesis, which helps maintain postprandial bile acid homeostasis [28,29]. This negative feedback regulation is important, as bile acid has strong and toxic detergent properties. FGF15/19 also regulates other postprandial responses, including the inhibition of gluconeogenesis, activation of glycogen and protein synthesis [30,31], and gallbladder filling [32]. The regulation of bile acid by FGF15/19 is dependent on βKlotho [33,34]. Interestingly, FGF15/19 and insulin (a representative postprandial hormone) share several postprandial effects, e.g., the induction of protein and glycogen synthesis [30]. Although plasma insulin levels are quickly elevated by feeding (within 1 h), the peak level of FGF15/19 in serum is achieved around 3 h after a meal [35], suggesting that FGF15/19 has a late-phase postprandial effect.

Feeding predominantly regulates the transcription of *FGF15/19*. Upon feeding, bile (containing bile acids) is released into the small intestine, which leads to the induction of *FGF15/19* expression via the nuclear bile acid receptor, farnesoid X receptor (FXR) [28,29,36]; this in turn regulates the transcription of target genes through interactions with the retinoid X receptor as a heterodimer. Other nuclear hormone receptors that are activated by bile acids, e.g., a vitamin D receptor, VDR [37] and a xenobiotic receptor pregnane X receptor [38], also regulate *FGF15/19*, indicating that they play an important role in bile acid metabolism. Bile acid is synthesized from cholesterol in the liver; sterol regulatory response element-binding protein 2, a master regulator of cholesterol synthesis, negatively regulates FGF15/19 expression through interactions with FXR [39]. Although studies have reported various regulation processes for *FGF15/19* transcription, little is known about the mechanism of FGF15/19 secretion. A recent study found that a natural genetic variant of the human *DIET1* gene increases the secretion of FGF15/19 in vitro [40,41].

### 2.2. FGF21

FGF21 was identified as a homolog of FGF15/19 [42], which is also a metabolic regulator in adipocytes [43]. FGF21 activates glucose uptake by upregulating the transcription of the glucose transporter *GLUT1*. *FGF21* is predominantly expressed in the liver and weakly expressed in white and brown adipose tissue [16,44]. Almost all of the circulating FGF21 in plasma is considered to be derived from the liver [45,46,47]. FGF21 produced in adipose tissue can function in an autocrine or a paracrine manner [48,49,50]. Released FGF21 works on target tissues, including adipose tissues, where the coreceptor βKlotho is expressed [11,51,52,53,54] (Figure 1). In adipose tissue, FGF21 stimulates a thermogenic response through the regulation of the *uncoupling protein 1* and *deiodinase-2* genes [52,55], and it activates peroxisome proliferator-activated receptor γ (PPARγ), a nuclear hormone receptor, by preventing its sumoylation. Such posttranslational regulation by FGF21 is important for the insulin-sensitizing effects of the drug thiazolidinedione, which is a chemical ligand for PPARγ [56]. Recent studies have also revealed that FGF21 can stimulate sympathetic nerve activity [57,58]. Interestingly, this nervous system activity is essential for the effects of FGF21 on energy metabolism [53,59]. FGF21 improves energy metabolism through multiple mechanisms. In adipose tissue, FGF21 activates fatty acid oxidation by upregulating lipolytic enzymes, hormone-sensitive lipase (HSL) and adipose triglyceride lipase (ATGL) [52,53,59,60]. FGF21 also activates nuclear fatty acid receptor PPARγ, a target of insulin sensitizer thiazolidinediones (TZDs), in adipose tissue. FGF21 prevents the sumoylation of PPARγ, resulting in an improvement in insulin sensitivity [56,61]. FGF21 reduces fatty liver by activating fatty acid β oxidation and reducing the expression of lipogenic genes [52,54,62,63]. FGF21 also regulates important fasting responses, including gluconeogenesis and ketone body synthesis [60,64,65,66]. During fasting, FGF21 induces the hepatic expression of *peroxisome proliferator-activated receptor coactivator protein-1α* (*PGC-1α*) to activate gluconeogenesis, and it increase ketogenesis by the induction of ketogenic enzymes *hydroxymethylglutaryl-CoA synthase 2* (*HMGCS2*) and *carnitine palmitoyl transferase 1a* (*CPT1a*). The other important roles of FGF21 include the regulation of growth [67], longevity [68], and pancreatic proteostasis [69]. The transgenic expression of FGF21 reduces size and growth by reducing growth hormone (GH) and the insulin-like growth factor-1 (IGF-1) signaling pathway through the decrease in the phosphorylation of signal transducer and activator of transcription 5 (STAT5) [67]. The other phenotype of FGF21 transgenic mice is an extension of lifespan through GH and IGF-1 signaling [68]. In an exocrine pancreas, where *FGF21* is highly expressed, FGF21 functions as a digestive enzyme secretagogue and maintains proteostasis during the postprandial state [69].

The expression of hepatic *FGF21* is strongly induced by fasting, which is the most well-known regulator of its transcription [60,65,70]. During fasting, free fatty acids travel from white adipose tissue (WAT) to the liver, where the nuclear fatty acid receptor PPARα activates the promoter activity of the *FGF21* gene by binding with peroxisome proliferator response elements comprising a direct repeat-1 element [60,65,70]. Other transcription factors that are activated during fasting, such as the glucocorticoid receptor (GR) [71,72] and cAMP responsive element-binding protein H (CREBH) [73,74], also regulate *FGF21* expression.

In addition to transcriptional regulation, it has been reported that circulating FGF21 protein has a short half-life [51,75,76]. Fibroblast activation protein (FAP), a serine protease, cleaves proteolytically and inactivates FGF21 [77,78,79,80]. As FAP protein is detected in human plasma, the pharmacological inhibition of FAP could be a therapeutic drug target for metabolic diseases, such as obesity and diabetes.

### 2.3. FGF23

The *FGF23* gene was first identified in a mutated form in patients with autosomal dominant hypophosphatemic rickets (ADHR) [81]. *FGF23* is highly expressed in osteocytes and osteoblasts; however, it is weakly expressed in the brain and thymus [16,82]. FGF23 is proteolytically processed to generate inactive fragments; however, some patients express FGF23 polypeptides which are resistant to proteolysis [81]. Moreover, excess circulating FGF23 can lead to hypophosphatemia and tumor-induced osteomalacia [83]. *FGF23* expression is regulated by vitamin D and dietary phosphate [84,85,86,87]. Transgenic mice overexpressing FGF23 exhibit hypophosphatemia, a decrease in circulating 1,25-dihydroxy vitamin D, and an increase in the renal release of phosphate [88,89,90]. FGF23-null mice and αKlotho-knockout mice share a similar phenotype, which includes increased levels of 1α-hydroxylase, an enzyme that functions in the production of active vitamin D [91,92,93]. This shared phenotype led to the association of the FGF23–αKlotho pathway. αKlotho is a type I transmembrane protein, and αKlotho-knockout mice exhibit an aging phenotype [93]. In contrast to FGF15/19 and FGF21, FGF23 requires αKlotho, which is predominantly expressed in the kidney, to bind to FGFR [10,12,94] (Figure 1). The FGF23–αKlotho pathway regulates phosphate excretion in the kidney and reduces the synthesis of vitamin D and parathyroid hormone (PTH). As an excess amount of circulating FGF23 is associated with ADHR/osteomalacia, several drug discovery studies have targeted FGF23 [19]. Currently, an antibody against FGF23 (burosumab) is used for patients with FGF23-related ADHR/osteomalacia, such as those with X-linked hypophosphatemia [95,96].

Vitamin D is a major regulator of *FGF23* transcription [87]. Indeed, 1α,25-dihydroxyvitamin D, an active form of vitamin D that is synthesized by 25-hydroxyvitamin D-1α-hydroxylase and expressed in the kidney, induces *FGF23* expression [97,98]. *FGF23* induction by 1α,25-dihydroxyvitamin D is mediated by the nuclear vitamin D receptor VDR [99,100]. PTH, a target molecule of the FGF23–αKlotho pathway, also regulates *FGF23* transcription through cAMP-dependent protein kinase A (PKA) and the Wnt pathway [101,102]. The PKA signal increases the mRNA expression of orphan nuclear receptor *Nurr1*, and then Nurr1 activates *FGF23* transcription [103]. In rickets model mice (Hyp, DMP1 knockout, and PHEX mutant), circulating FGF23 levels are substantially increased [104,105,106,107,108]. In these models, the activation of the nuclear factor of activated T-cells (NFAT) contributes to *FGF23* induction [109,110].

## 3. Stress Signaling and Endocrine FGFs

### 3.1. ER Stress

The ER is the subcellular organelle responsible for many cellular functions, including the folding and maturation of proteins, the synthesis of lipids, and the regulation of calcium storage in eukaryotic cells. To maintain proper protein folding in the ER, various ER-resident molecular chaperones assist with ER quality control. BiP/GRP78 (immunoglobulin heavy chain binding protein) is the major molecular chaperone in the ER; it is an important regulator of the unfolded protein response pathway. When ER homeostasis is dysregulated by pathological and pharmacological conditions, unfolded or misfolded proteins accumulate in the ER lumen; this is referred to as ER stress. Upon ER stress, three branches of unfolded protein response (UPR) are activated to restore ER homeostasis. In mammals, the UPR pathway adjusts the protein folding capacity in the ER by activating three ER-resident transmembrane sensors, namely, inositol requiring enzyme 1 (IRE1), protein kinase RNA-like ER kinase (PERK), and activating transcription factor 6 (ATF6) [111] (Figure 2). IRE1, a type I transmembrane protein, was first identified as a UPR transducer in yeast [112,113]. Mammals express two IRE1 homologs: IRE1α and IRE1β [114,115]. *IRE1α* is ubiquitously expressed and functions as an ER stress transducer, whereas the expression of *IRE1β* is restricted to the gut. Upon ER stress, IRE1 is oligomerized to activate its RNase domain. IRE1 cleaves a substrate mRNA from *Hac1* in yeast and *XBP1* in mammals, respectively. After unconventional splicing by IRE1, the mature mRNA of *Hac1* and *XBP1* is translated to produce a basic leucine zipper-type transcriptional factor. PERK, another UPR transducer, is also a type I transmembrane protein that resembles IRE1 [116,117,118]. In the presence of ER stress, PERK is oligomerized for the activation of autophosphorylation. In contrast to IRE1, active PERK phosphorylates the α subunit of eukaryotic translation initiation factor 2 (eIF2α), which leads to the inhibition of global translation and protein loading to the ER. In contrast, PERK selectively activates the translation of mRNA that encodes *ATF4*, a basic leucine zipper-type transcriptional factor. ATF4 regulates the expression of genes related to amino acid metabolism and antioxidative responses [119]. ATF6, a type II transmembrane protein, is another UPR transducer. In mammals, ATF6 has two subtypes (ATF6α and ATF6β) that are ubiquitously expressed. Upon ER stress, ATF6 transits from the ER to the Golgi, where it receives two-step proteolysis by site-1 and site-2 proteases [120]. This cleavage results in the release of the N-terminal transcription factor domain (ATF6-N). In the nucleus, ATF6-N regulates transcription related to ER chaperones and ER-associated degradation [121].

Several studies have revealed that the transcriptional regulation of endocrine FGFs occurs with ER stress (Figure 2, Table 1). In addition to bile acids, ER stress is also a regulator of *FGF15/19* transcription [122]. *FGF15/19* is a direct target gene of ATF4, which is stimulated by ER stress. ATF4 is also known to bind to the *FGF15/19* promoter through an amino acid response element (AARE) upon ER stress. As ER stress is triggered by high concentrations of bile acids [123,124], the ATF4–FGF15/19 pathway may have a function in preventing the toxicity that can be induced by excess bile acids. In addition to ER stress, ATF4 activation is regulated by various stress signaling pathways, such as oxidative stress and amino acid deprivation [111]. We previously found that *FGF15/19* is selectively regulated by ER stress; however, it is not regulated by other ATF4 stimuli [125].

In addition to *FGF15/19*, ATF4 is reportedly an important regulator of *FGF21* transcription [125,126,127,129,130]. Interestingly, the *FGF21* promoter contains three AAREs, which provide potent induction by ER stress [150]. We found that *FGF21* expression is induced by ER stress, oxidative stress, and amino acid deprivation, whereas *FGF15/19* expression is ER stress selective [125]. Another ER stress-activated transcription factor, X box–binding protein-1 (XBP-1), has also been reported to regulate *FGF21* expression; however, further research is required in this area as the binding element of XBP-1 is not conserved among species and XBP-1 failed to activate the human *FGF21* promoter [150].

In a hepatocellular carcinoma cell line, FGF15/19 reduces ER stress through the activation of the antioxidative transcription factor nuclear factor erythroid 2-related factor 2 (NRF2) [151]. The overexpression of FGF15/19 increases the phosphorylation of glycogen synthase kinase-3β (GSK3β), leading to the inhibition of the proteasomal degradation of NRF2.

In contrast to FGF15/19, several studies have described that secreted FGF21 inhibits ER stress [128,152,153,154]. FGF21 alleviates drug-induced ER stress through MAP kinase [128,154]. FGF21 also reduced ER stress-induced steatosis [128]. ER stress is also triggered by physiological conditions, such as the postprandial state and a secretagogue response in the pancreas [155,156,157]. We previously reported that FGF21 overexpression by adenovirus is effective in reducing refeeding-induced ER stress [152]. In skeletal muscle, although the basal expression of *FGF21* is low [16], *FGF21* expression is strongly induced by the forced activation of PERK; this can help prevent obesity [158].

Unlike *FGF15/19* and *FGF21*, the regulation of *FGF23* by ER stress has not been reported. We observed that two ER stress inducers, tunicamycin and thapsigargin, fail to increase the gene expression of *FGF23* in osteoblasts (Shimizu et al., unpublished observation). Thus, ER stress seems to selectively regulate *FGF15/19* and *FGF21* but not *FGF23* (Figure 2).

### 3.2. Oxidative Stress

Oxidative stress is triggered by the disruption of the balance between the production of reactive oxygen species and antioxidants. Excess oxidative stress causes oxidative damage to cellular components, including proteins, DNA, and lipids.

NRF2 is a key transcriptional regulator of antioxidant responses; it regulates the expression of the phase II detoxifying enzyme and antioxidant-responsive genes by binding to antioxidant-responsive elements that are present in the promoter regions of target genes [159,160]. Under unstressed conditions, NRF2 interacts with Kelch-like ECH-associated protein 1 (Keap1), an actin-binding cytoplasmic protein, to repress NRF2 activity through degradation by cullin 3 (Cul3) ubiquitin E3 ligase [161]. However, oxidative stress triggers the oxidation of the cysteine residues of Keap1, which leads to its conformational change and the liberation of NRF2 to the nucleus for the induction of its target genes (Figure 3).

ATF4 is also a transcriptional regulator of oxidative stress. Unlike ER stress, oxidative stress increases ATF4 translation through the activation of heme-regulated inhibitor (HRI), another eIF2α kinase. HRI is also stimulated by various stresses, including heme deprivation and iron deficiency [162]. Similar to PERK, HRI is activated by its autophosphorylation, which leads to the phosphorylation of eIF2α and an increase in ATF4 translation (Figure 3).

*FGF15/19* expression is increased by oxidative stress in intestinal cell lines; however, oxidative stress fails to increase *FGF15/19* expression in vivo, whereas other ATF4 target genes are induced [125] (Figure 3, Table 1). As *HRI* and *ATF4* are expressed in the intestine, the mechanism of *FGF15/19* selective regulation is currently unclear. In human hepatocytes that express *FGF15/19*, oxidative stress increases the expression of *FGF15/19*, suggesting a conserved regulation by oxidative stress at least in vitro [151]. During the postprandial state, secreted FGF15/19 increases the phosphorylation of GSK3β, an inactive form of GSK3β, which stimulates glycogen synthesis [30]. Interestingly, the stability of the NRF2 protein is regulated by GSK3β [163,164]. Consistent with this phenomenon, FGF15/19 activates the NRF2 pathway by inactivating GSK3β in hepatocytes and cardiomyocytes, which helps protect cells and tissues [151,165,166] (Figure 3).

In contrast to *FGF15/19*, oxidative stress increases the expression of *FGF21* both in vitro and in vivo [125,126] (Figure 3, Table 1). Although ATF4 induces *FGF21* expression, several studies have reported negative regulation by NRF2 [134,136,137,138]. NRF2 is also reported to activate *FGF21* expression in diabetes [135]. Several studies have found that oxidative stress is reduced by FGF21 [167,168,169,170] (Figure 3). In human umbilical vein endothelial cells, FGF21 prevents hydrogen peroxide-induced oxidative damage and cytotoxicity by affecting stress-responsive kinases, including p38 MAP kinase and JNK [170]. In the liver, FGF21 reduces acetaminophen-induced oxidative stress through an increase in NRF2 expression. A transcriptional coactivator, PGC-1α [167], mediates NRF2 induction by FGF21. The reduction in oxidative stress due to this pathway is lost in FGF21-knockout mice, which indicates the physiological importance of this pathway. In cardiomyocytes, FGF21 prevents cardiac hypertrophy by reducing oxidative stress [168]. FGF21 also induces the expression of antioxidant proteins, including *superoxide dismutase* (*SOD*) *2* and *uncoupling protein 3*, but such effects are not observed in FGF21-knockout mice or following treatment with FGF21 antibody. Circulating FGF21 levels are known to be increased in patients with rheumatoid arthritis [171]. When FGF21 is administrated to rheumatoid arthritis model mice, the levels of some antioxidant proteins, including SODs, increase, which in turn reduces oxidative stress and inflammation [169].

Although the regulation of *FGF23* expression by oxidative stress has not been reported, FGF23 is known to activate NRF2 signaling in osteoblasts [172]. Dexamethasone (DEX), a synthetic glucocorticoid, is used for patients with a chronic inflammatory disease. DEX is known to induce reactive oxygen species in osteoblasts [173,174], and DEX-induced osteoporosis is a major side effect [175]. The treatment of osteoblasts with FGF23 increases NRF2 protein levels through the FGFR1–Akt pathway and reduces oxidative stress, which in turn protects against DEX-induced cytotoxicity [172] (Figure 3). Interestingly, the FGFR1–Akt pathway can be activated in the absence of αKlotho [156]. As the medical use of NRF2 activators at high concentrations is limited due to side effects, FGF23 is an attractive target for therapies.

### 3.3. Mitochondrial Stress

The mitochondria, an organelle with a double membrane and unique circular DNA, has many important functions, including ATP synthesis. Mitochondrial dysfunction caused by metabolic changes within mitochondria and the disruption of mitochondrial quality control results in mitochondrial stress. Unlike that of ER stress, the precise mechanism of the mitochondrial stress response is not well characterized. However, recent findings indicate that mitochondrial stress affects several metabolic pathways and diseases [176].

The basal expression of *FGF21* is low in skeletal muscle [16], but FGF21 is now recognized as a myokine, i.e., a protein produced and released from muscle fibers [177,178]. Mitochondrial dynamics (mitochondrial fusion and fission) are important for maintaining mitochondrial function. The deficiency of optic atrophy 1 (OPA1), an essential protein for mitochondrial fusion, causes the potent induction of *FGF21* in skeletal muscle [139]. The ablation of OPA1 in skeletal muscle causes mitochondrial stress response, which increases the expression of *ATF4* and *FGF21* (Figure 4). In OPA1/FGF21 double-knockout mice, muscle atrophy caused by OPA1 deficiency is partially recovered. Thus, skeletal muscle-derived FGF21 apparently functions in an autocrine manner. Although βKlotho can be detected in skeletal muscle [139], it is expressed at low levels [16]. Recombinant FGF21 treatment fails to increase the phosphorylation of ERK1/2, a target of FGFR substrate 2 (FRS2), which is activated by FGF21 signaling, in skeletal muscle [13]. Thus, further studies may be required to confirm the precise functional mechanism of FGF21 in skeletal muscle. The disruption of autophagy-related gene 7 (ATG7), an important factor for autophagosome expansion and completion, in skeletal muscle results in autophagy deficiency. The resultant mitochondrial stress, as well as the inhibition of the mitochondrial respiratory chain, leads to the induction of *FGF21* through the eIF2α–ATF4 pathway [130]. This skeletal muscle-specific deletion of ATG7 increases energy expenditure and prevents diet-induced obesity and the amelioration of insulin resistance by the activation of lipolysis and the browning of white adipose tissue. Both ATF7 and FGF21 deficiency diminish these metabolic changes, indicating their physiological importance. *FGF21* induction by mitochondrial stress is also observed in other mitochondrial dysfunctions, including mitochondrial myopathy and mutations of mitochondrial DNA [179,180]. Therefore, mitochondrial stress appears to induce *FGF21* expression to protect against metabolic abnormalities. Although *FGF21* induction contributes to muscle atrophy and production of inflammatory cytokines [139], FGF21 activates mitophagy to degrade dysfunctional mitochondria [181]. Thus, further studies are required to fully understand the function of FGF21 on mitochondrial dysfunctions (Figure 4).

Unlike *FGF21*, the regulation of *FGF15/19* and *FGF23* by mitochondrial stress has not been reported. FGF15/19 is, however, known to alleviate mitochondrial dysfunction through the AMPK–PGC-1α pathway [182] (Figure 4).

### 3.4. Thermal (Cold) Stress

In humans, the thermoregulation system maintains the core body temperature at around 37 °C. Whereas white adipose tissue stores chemical energy as triglycerides, brown adipose tissue is a specialized tissue that dissipates chemical energy to produce heat in a process known as nonshivering thermogenesis [183,184]. In addition to brown adipose tissue, other types of thermogenic adipocytes, termed beige or brite adipocytes, are known to exist. In response to acute cold stress, *uncoupling protein 1* (*UCP1*), the mitochondrial uncoupling protein, is potently induced in both brown adipose tissue and beige adipocytes by a set of transcription factors, including PGC-1α and ATF2 [183]. UCP1 uncouples electron transport from ATP synthesis by dissipating the mitochondrial proton motive force (Δp) and thereby increases thermogenesis. As brown adipose tissue and beige adipocytes consume triglycerides, these tissues may be attractive targets for the treatment of obesity and type 2 diabetes [184].

*FGF21*, but not *FGF15/19* or *FGF23*, is expressed in white adipose tissue and brown adipose tissue [16]. Upon cold stress, *FGF21* is strongly induced in these tissues independent of PPARα, a key regulator of hepatic *FGF21* [55,140]. Instead, p38 MAPK-mediated ATF2 activation is important for *FGF21* induction during cold stress [55] (Figure 5). Interestingly, G protein-coupled receptor 120, which is activated by long chain fatty acids, stimulates the release of FGF21 from adipocytes [185]. The levels of circulating FGF21 are almost abolished by the liver-specific deletion of FGF21, whereas they are unchanged by adipose-specific deficiency. Thus, FGF21 produced in brown adipose tissue and white adipose tissue is thought to function in an autocrine and a paracrine fashion but not an endocrine fashion, unlike that produced in the liver [49,56]. Chronic FGF21 treatment activates the thermogenic response of adipocytes [49,186] at least partly through the increase in thermogenic coactivator PGC-1α protein levels and the subsequent increase in *UCP1* expression [49]. This thermogenic activation by FGF21 is also observed in human neck-derived primary adipocytes [187]. Thus, FGF21 is a cold stress-induced adipokine, and it activates the thermogenic response to protect against further cold stress (Figure 5). Although the regulation of *FGF15/19* by cold stress has not been reported, circulating FGF15/19 levels positively correlate with *UCP1* expression [188]. The overexpression of FGF15/19 induces the expression of thermogenic genes, including *UCP1* and *PGC-1α* in subcutaneous WAT, whereas FGF15/19 deficiency prevents this induction. Thus, FGF15/19 activates the thermogenic response through a browning of WAT (Figure 5).

### 3.5. Nutrient Stress

The expression of *FGF15/19* and *FGF21* is observed in tissues that are important for nutrient sensing, including the intestine, liver, and adipose tissue [16]. In these tissues, signaling pathways for nutrient stress are activated in response to severe nutritional states, including nutrient deficiency or overnutrition. Nutrient deficiency, fasting, or over nutrition elicit nutritional stress signals and compensatory survival mechanisms. *FGF15/19* and *FGF21* are important responsive genes for feeding or fasting [29,60,65]. During fasting, the expression of *FGF21* is regulated by several transcription factors, including PPARα, GR, and CREBH [60,65,70,71,72,73,74]. In addition, the overconsumption or deficiency of each major macronutrient (e.g., amino acids, lipids, and carbohydrates) triggers nutrient stress signaling (Figure 6).

Amino acid deprivation or protein restriction are known to activate transcription factor ATF4. Unlike during ER stress and oxidative stress, general control non-derepressible 2 (GCN2) phosphorylates eIF2α, leading to an increase in ATF4 translation [189]. Although both ATF4 and GCN2 are expressed in the intestine, we did not observe a significant change in *FGF15/19* expression under a leucine-deficient diet [125]. In contrast, hepatic *FGF21* is reportedly induced upon amino acid deprivation and protein restriction both in vitro and in vivo [125,129,132,133]. During amino acid or protein restriction, induced FGF21 reduces the size of adipocytes through the activation of lipolysis, and it activates thermogenesis through the induction of *UCP1* in brown adipose tissue [132,190]. We previously reported that *FGF21* expression is induced by β-conglycinin, a soy protein [191]; the administration of β-conglycinin to mice results in a methionine imbalance in the portal vein, which in turn activates the ATF4–FGF21 pathway. FGF21 deficiency prevents β-conglycinin-induced improvements in energy metabolism, including the reduction in body weight gain and adipose tissue weight.

In addition to the effects of feeding and bile acids, *FGF15/19* expression is increased by saturated fatty acids, which cause lipotoxicity and ER stress [122,192,193]. In obese patients, circulating FGF15/19 is known to be decreased [192,194]. Furthermore, FGF15/19 prevents hepatic steatosis and reduces hepatic ER stress in high-fat-diet fed mice [192,195,196].

In contrast to the effect of nutrient deficiency [60,65,70,71,72,73,74], *FGF21* expression is increased by overnutrition. For example, under a high-carbohydrate diet, hepatic *FGF21* is strongly induced through the activation of carbohydrate response element-binding protein (ChREBP) [141,142,143,144,145,146,147]. The overexpression of ChREBP induces *FGF21* expression and improves glucose tolerance and plasma triglyceride, despite the occurrence of fatty liver [197,198]. FGF21 can also help decrease sugar intake and preference [199,200,201], suggesting that a negative feedback loop regulates sugar consumption via the ChREBP–FGF21 pathway. The expression of hepatic *FGF21* is also increased in obese mice, such as ob/ob mice, and under high-fat diet and fatty liver conditions [44,70,202,203,204,205,206,207,208,209], in which FGF21 improves energy metabolism. In contrast to the liver, *FGF21* in WAT is induced by feeding, which is likely mediated by PPARγ [56,148,149]. The expression of *FGF21* in WAT is also increased during obesity [148] when adipose PPARγ is activated to promote adipogenesis and lipid accumulation [210]. Thiazolidinediones (TZD), an antidiabetic PPARγ ligand, have been reported to improve insulin sensitivity through *FGF21* induction [56]. FGF21 increases PPARγ activity through the prevention of its sumoylation. The effects of FGF21 on WAT are likely mediated in an autocrine or paracrine fashion, but not via endocrine action [45,56].

## 4. Conclusions

Several studies have indicated that endocrine FGFs, especially FGF15/19 and FGF21, are attractive therapeutic target molecules for the treatment of metabolic disorders. Both FGF15/19 and FGF21 activate energy expenditure and reduce body weight gain despite their transcriptional regulator and tissue expression patterns being different. Cellular stress and energy metabolism are closely related. For example, ER stress not only disrupts ER homeostasis due to protein folding, but also has effects on obesity and type 2 diabetes [211]; moreover, reducing ER stress improves energy metabolism [212]. An imbalance between oxidants and antioxidant systems can lead to a variety of diseases, including type 2 diabetes and atherosclerosis. Among the endocrine FGFs, *FGF15/19* and *FGF21* are selectively responsive to stress signaling. In particular, FGF21 is described as a “stress hormone” as it is strongly induced by various stress signals [213]. ATF4 is likely a key regulator of *FGF15/19* and *FGF21* during stress signaling. Endocrine FGFs have protective effects against cellular stresses in addition to improving energy metabolism. As both *FGF15/19* and *FGF21* are target genes of ATF4, it would be interesting to develop an ATF4 activator as an inducer of *FGF15/19* and *FGF21* expression. The soy protein β-conglycinin is a good example of an ATF4 activator that can prevent metabolic disorders through the induction of *FGF21* without ER stress [191]. In conclusion, endocrine FGFs, especially FGF15/19 and FGF21, play important roles in stress signaling. However, further studies on the role of stress–endocrine FGF pathways are required to evaluate potential therapeutic targets for stress toxicity and metabolic disorders.

## Figures and Tables

**Figure 1 cells-11-00505-f001:**
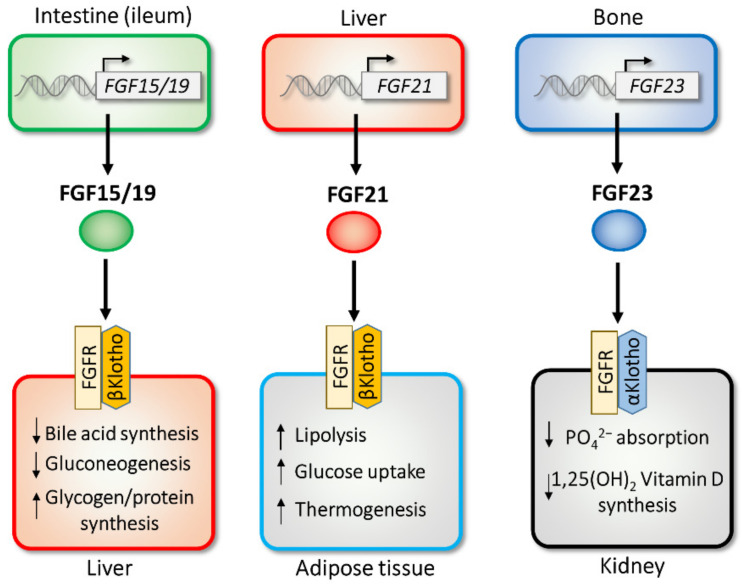
Endocrine functions of FGF15/19, FGF21, and FGF23. *FGF15/19*, *FGF21*, and *FGF23* are expressed in intestine, liver, and bone, respectively. Secreted FGFs selectively act on the target tissues through FGF receptor (FGFR)/Klotho receptor complexes.

**Figure 2 cells-11-00505-f002:**
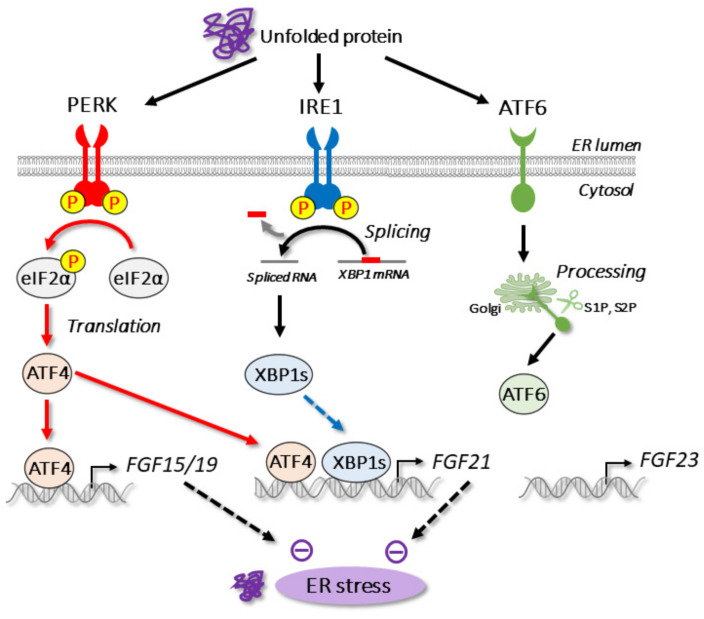
ER stress and endocrine FGFs. Upon ER stress, three branches (PERK, IRE1, and ATF6) are activated to maintain ER homeostasis. Expression of *FGF15/19* and *FGF21* genes, but not *FGF23* gene, is regulated by ER stress.

**Figure 3 cells-11-00505-f003:**
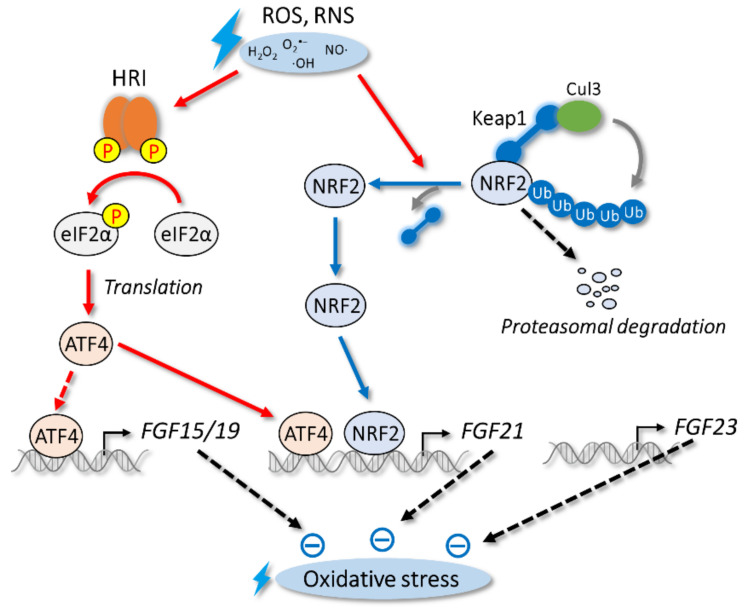
Oxidative stress and endocrine FGFs. HRI–ATF4 and Keap1–NRF2 pathways are activated in response to oxidative stress. Expression of *FGF15/19* and *FGF21* genes, but not *FGF23* gene, is regulated by oxidative stress.

**Figure 4 cells-11-00505-f004:**
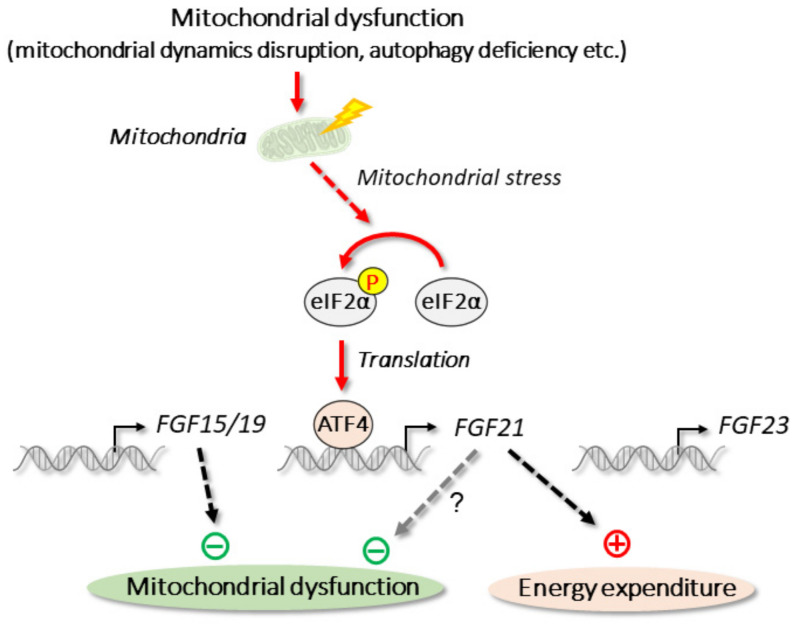
Mitochondrial stress and endocrine FGFs. Mitochondrial stress is triggered by mitochondrial dysfunctions or by deficiency of mitochondrial dynamics-related gene *OPA1* or autophagy-related gene *ATG7*. In skeletal muscle, mitochondrial stress induces *FGF21* through eIF2α–ATF4 pathway.

**Figure 5 cells-11-00505-f005:**
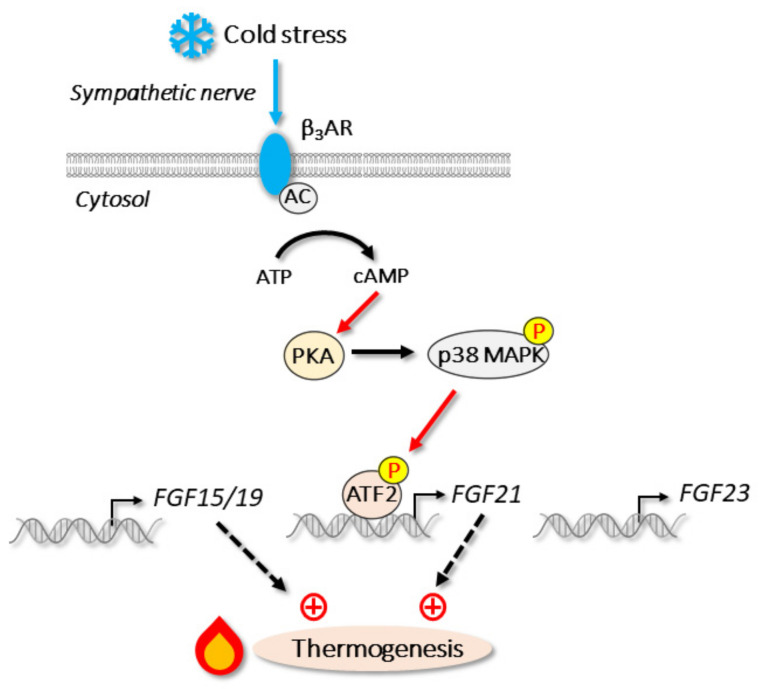
Cold stress and endocrine FGFs: upon cold stress, β_3_-adrenergic receptor (β_3_AR) stimulates the production of second messenger cAMP through adenylate cyclase (AC). Subsequent pathway actives p38 MAPK and transcription factor ATF2, which induces *FGF21* expression during cold stress.

**Figure 6 cells-11-00505-f006:**
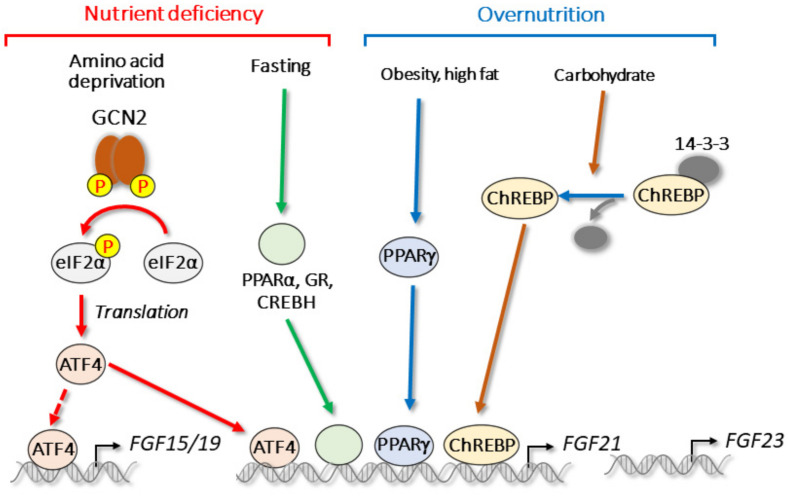
Nutrient stress and endocrine FGFs. Both nutrient deficiency and overnutrition regulate endocrine FGFs.

**Table 1 cells-11-00505-t001:** Regulation of endocrine FGFs by stress signaling.

FGFs	Regulator	Stimuli	Refs
*FGF15/19*	ATF4	ER stress	[122]
	ATF4 ^1^	Oxidative stress	[125]
	ATF4 ^1^	Amino acid deprivation	[125]
*FGF21*	ATF4	ER stress	[125,126,127]
	XBP-1	ER stress	[128]
	ATF4	Oxidative stress	[125,126]
	ATF4	Amino acid deprivation	[125,129,130,131,132,133]
	NRF2 ^2^	Oxidative stress	[134,135,136,137,138]
	ATF4	Mitochondrial stress	[130,139]
	ATF2	Cold stress	[55,140]
	PPARα	Fasting	[60,65,70]
	GR	Fasting	[71,72]
	CREBH	Fasting	[73,74]
	ChREBP	High carbohydrate	[141,142,143,144,145,146,147]
	PPARγ	Obesity/feeding	[56,148,149]
*FGF23*	-	-	-

^1^ Only in vitro. ^2^ Both positive and negative regulations have been reported.

## Data Availability

Not applicable.

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
