# Peer review of "Endocrine Fibroblast Growth Factors in Relation to Stress Signaling"

_cells, 2022, doi:10.3390/cells11030505_

Round 1

Reviewer 1 Report

The review by Shimizu and Sato is well written, comprehensive and nicely structured manuscript. It provides a thorough background of FGF signaling in response to physiological stress.

Specific comments : 

Authors should discuss about the structure/biology of FGFs more elaborately in the introduction section. The different type of FGFs in drosophila, mouse and humans should be discussed. Also the major signaling pathways under the regulation of FGFs should be discussed.

Please check line 156. 

Please check line 327-328 for punctuation. 

Reviewer 2 Report

The manuscript by Shimizu and Sato is a concise, yet comprehensive review on the role and regulation of endocrine FGFs by stress signaling. The endocrine FGF research field is still relatively new, but recent advances in the field have opened exciting new avenue for development of new medical strategies to combat metabolic abnormalities. Hence, this review is very timely. While the field of endocrine FGFs has been previously reviewed, the specific objective of Shimizu and Sato's manuscript was to review endocrine FGFs in the context of different cellular stresses. This review does accomplish that and presents a significant contribution to the field. Moreover, the review is well-written and well-structured, which makes it easy to read and highly accessible. Hence, I gladly recommend the manuscript for publication in cells. My minor comments below are largely cosmetic in nature. I recommend careful proof-reading of the manuscript since I detected several typos (some of which are listed below). 

1) Readers who may want to dive even deeper into endocrine FGF biology may benefit from an additional reference to an excellent review by Phan et al., The Saga of Endocrine FGFs, which is a deep dive into the history of endocrine FGFs and endocrine FGF structure.

2) On page 3, ll. 129-130, it would be beneficial to clarify that these are not any missense mutations but those found in ADHR patients.

3) On page 6, l. 213, I assume that the authors mean "to regulate FGF21 expression"?

4) A few of the typos I detected include:

page 5, l. 186, UPR

p. 7 l. 324, also

p. 7 l. 329, degrade

p. 10 l. 365, regulation

Reviewer 3 Report

The is a generally well organized and comprehensive review of endocrine fibroblast growth factors. It it timely and should be of general interest. I have a list of generally minor concerns:

  1. The review focuses on endocrine FGFs, and this should be included in the title. As is, the title is not reflective of the content  
  2. Some sentences/statements lack clarity and should be expanded upon:
  3. Lines 11-113, please break in individual components (fatty acid oxidation; insulin sensitivity;fatty liver)and expand briefly on the action of FGF21.
  4. Similarly, expand on what regulation by FGF21 means in lines 114-115.
  5. Also, lines 122-124, what does short half life refer to, and briefly explain the protease action (line 124).
  6. Lines 127-130. Expand briefly: what is the significance of cleavage to N- and C-fragments in terms of action.
  7. Lines 183-185.  Why "paradoxically"? ATF4 is also important for lysosomal biogenesis, which is worth mentioning.
  8. Legend in Figure 2. The second sentence is incomplete. "regulated by ???"
  9. Lines 219-222. And 231-234. The authors may consider toning down reliance on their unpublished data. 
  10. Sentence starting in line 224 does not make clear sense
  11. Line 237 lacks clarity as written. One suggestion would be to rephrase as "Oxidative stress is triggered by the disruption of the balance between production of reactive oxygen species and antioxidants"
  12. Lines 245-247. Expand briefly on the Keap1-NRF2 relationship
  13. Figure 3, legend, line 251. Replace "regulated" with "unregulated"
  14. Lines 261-262. Meaning unclear.
  15. Lines 287-289. Clarify how are the two parts of the sentence connected to each other
  16. Lines 313-316. Since growth factors activate a multitude of pathways, reliance on only ERK1 needs better justification
  17. Lines 331-333. Again, please explain the connection between the two statements in the sentence
  18. Figure 4. Consider adding genetic defects as contributors to mitochondrial dysfunction
  19. Lines 357-359. Expand and clarify
  20. Lines 379-380. The sentence lacks clear meaning as written. Do the authors mean  "Nutrient deficiency, fasting, or over nutrition elicit nutritional stress signals and compensatory survival mechanisms"
